# Evaluation of the Milling Accuracy of Zirconia-Reinforced Lithium Silicate Crowns Fabricated Using the Dental Medical Device System: A Three-Dimensional Analysis

**DOI:** 10.3390/ma13204680

**Published:** 2020-10-21

**Authors:** Seen-Young Kang, Ji-Min Yu, Jun-Seok Lee, Ki-Sook Park, Seung-Youl Lee

**Affiliations:** Medical Device Research Division, National Institute of Food and Drug Safety Evaluation, Chungcheong buk-do 28159, Korea; seenyoung@korea.kr (S.-Y.K.); yjm9223@korea.kr (J.-M.Y.); junseok1025@korea.kr (J.-S.L.); kispark@korea.kr (K.-S.P.)

**Keywords:** dental medical system, accuracy, zirconia-reinforced silicate

## Abstract

This study aimed to analyze the milling accuracy of lithium disilicate and zirconia-reinforced silicate crown fabricated using chairside computer-aided design/manufacturing (CAD/CAM) system. Mandibular left first premolar was selected for abutment. A master model was obtained for digital impression using an intraoral scanner, and crowns were designed using a CAD software design program. Amber Mill (AM), IPS e max CAD (IPS), and CELTRA DUO (CEL) were used in the CAD/CAM system, and a total 45 crowns (15 crowns each for AM, IPS, and CEL) was fabricated. Milling accuracy was analyzed with respect to trueness, measured by superimposing CAD design data and scan data through a three-dimensional program to compare the outer and inner surfaces and internal and external parts, thereby acquiring both quantitative and qualitative data. Data were analyzed using the non-parametric test and Kruskal–Wallis H test. In addition, the Mann–Whitney U test was used by applying the level of significance (0.05/3 = 0.016) adjusted by post-analysis Bonferroni correction. All the measured parts of the lithium disilicate and zirconia-reinforced silicate crowns showed statistically significant differences (*p* < 0.05). The lithium disilicate (AM and IPS) materials showed superior milling accuracy than the zirconia-reinforced lithium silicate (CEL) materials.

## 1. Introduction

In the field of dentistry, the use of functional recovery and oral aesthetic treatments employing ceramic materials has been increasing [1,2]. Generally, aesthetic dental prostheses are made by fusing porcelain to metal ceramic prostheses or through heat-pressing of glass-ceramic materials [3,4]. Conventionally, a wax pattern is fabricated using the patient’s work model, followed by the preparation of a metal coping after investment and casting processes. The aesthetic prostheses are formed through ceramic build-up or thermal press molding methods using a ceramic ingot, which require complicated processing procedures and time [3,4,5].

Recently, the demand for the preparation of ceramic material-based aesthetic prostheses using the dental computer-aided design/manufacturing (CAD/CAM) system has increased significantly [6]. In particular, the CAD/CAM system has the advantage of reducing the manufacturing errors by decreasing the patient’s operation time and labor and eliminating complicated processing procedures that occur in the laboratory [7]. Accordingly, various ceramic materials for dental CAD/CAM have been introduced. Lithium disilicate glass ceramic has recently been adopted into the CAD/CAM system [8], and it is widely used as an aesthetic material owing to its excellent aesthetics, mechanical features, adhesive cementation, and adequate marginal fit among crystallized glass ceramics [9,10].

However, CAD/CAM ceramic prostheses are produced using the wet-grinding process. Due to the inherent brittleness of ceramics, this process is vulnerable to cracks or fractures [11,12]. When these prostheses are used for fabricating ceramic restoration, there are chances of clinical failure due to the absorption of occlusal force. Furthermore, dental caries and periodontal disease can be triggered due to prosthesis fracture, microleakage, or external bacterial invasion [13,14,15]. Indeed, it is important to manufacture a perfect prosthesis in order to avoid problems that may occur during the restoration of the dental prosthesis in the oral cavity. However, the most reliable method to protect the teeth is to prevent the initial formation of cavities in the oral cavity [16].

Novel materials mixed with glass ceramic and zirconia were recently introduced to minimize the errors during the production of CAD/CAM ceramic crowns. When 10% zirconium oxide was included in a zirconia-reinforced lithium silicate ceramic, the processing errors during the production of CAD/CAM ceramic crowns were reduced, and mechanical features and aesthetics of the glass ceramic were complemented [17]. Particularly, the zirconium oxide particles prevented cracking and strengthened the ceramics. Due to this addition, physically enhanced and aesthetically superior aesthetic restorations can be fabricated [18].

Zirconia possesses chemical and volume stability, with higher bending strength as well as fracture toughness when compared with the previously reported ceramic materials [18]. However, it has few limitations, such as abrasion of the antagonists triggered by its high strength, difficulty to reproduce colors, and weak adhesion with resin cements as it is opaquely white [19,20,21]. Combining zirconia with glass ceramic increases the strength of the ceramic and adhesion, in addition to forming aesthetic and excellent resin cements. Since zirconia-reinforced silicate ceramic is fully crystallized, the errors from prosthesis contraction due to heat treatment can be controlled [22,23].

Milling accuracy is significant from the viewpoint of fabricating ceramic prostheses. Low milling accuracy can cause errors during the fabrication of dental prostheses, leading to clinical failure and marginal chipping [24,25]. Particularly, CAD/CAM dental ceramics undergo the process of prosthesis through the milling process, and measuring prosthesis error is very limited, according to the type of ceramic. Moreover, there has been little research conducted on this topic. Therefore, this study aimed to analyze and evaluate milling errors according to the types of ceramics through qualitative and quantitative analysis. A three-dimensional (3D) measuring program was used for the analysis, which is frequently adopted for measuring scanner errors and analyzing mechanical errors [26,27,28], in addition to assessing the accuracy of prostheses [29,30].

The null hypothesis of this study is that there is no significant difference between the milling accuracy of lithium disilicate and zirconia-reinforced silicate crown fabricated using the dental CAD/CAM system.

## 2. Materials and Methods

### 2.1. Study Design and Experimental Condition

The flow chart of this protocol is shown in Figure 1.

The experiment was conducted at a temperature of 23 ± 1 °C, in accordance with ISO 554 [31]. To maintain consistent experimental conditions, identical crown STL files were used for each material. For grinding burs, cylinder pointed bur 12S and step bur 12S were used. Fifteen crowns were fabricated, and then fabrication errors were removed, according to bur usage, by replacing the grinding bur.

### 2.2. Selection of the Master Model and Acquisition of Impression Scanning Data

The mandibular master model (AG3; Frasaco GmbH, Tettnang, Germany) and mandibular left first premolar (AG-3 ZPVK 44; Frasaco GmbH, Tettnang, Germany) abutment were selected as the materials. Digital impressions were obtained for the master model and abutment with an intraoral scanner (CEREC Omnicam, Sirona Dental Systems, Bensheim, Germany).

### 2.3. Fabrication of the Crown Using the Dental Milling Machine

Impression scan data were transported into a design software program (CEREC inLab software v4.2; Sirona Dental Systems GmbH, Bensheim, Germany) and then used to design the crowns. Crowns were set to cement space = 80 μm, occlusal milling offset = 125 μm, contact strength = 25 μm, and occlusal strength = 0 μm, while the most clinical library crown was applied, and the CAD Design STL file was completed. The completed STL file was transferred to a milling machine (inLab MCXL; Sirona Dental Systems GmbH, Bensheim, Germany) and 45 ceramic crowns, i.e., 15 crowns for each material, were fabricated (Table 1). Three groups were formed as follows: AM (Amber Mill), IPS (IPS e.max CAD), and CEL groups (CELTRA DUO).

After milling was completed, completed ceramic crowns attached to the ceramic blank were separated by removing the connected holder, The remaining parts were removed using a diamond bur grinder. Then, the ceramic crowns underwent heat treatment, in accordance with the manufacturer’s instructions, to complete the crystallization process (Table 2).

### 2.4. Scanning of the Outer and Inner Surfaces of Crowns

Scan spray (CerecOptispray, Sirona Dental Systems, Bensheim, Germany) was applied on the outer and inner surfaces of the completed ceramic crowns in accordance with the manufacturer’s instructions. Furthermore, scan data were obtained using a lab scanner (Identica blue, Medit, Seoul, Korea). Unnecessary parts were removed after referencing data were acquired by a 3D program (Geomagic Control 2015, Geomagic GmbH, Rock Hill, SC, USA). Additionally, the inner surface was divided into internal and external parts, as per the criterion of marginal above 1 mm for in-depth analysis.

### 2.5. Three-Dimensional Accuracy Analysis

Each measured part (outer and inner surfaces and external and internal parts) was used to align the reference data with the scan data for best fit by superimposing and assessing them (Figure 2).

To identify qualitative variations, max/min tolerance was set to ±50 μm and max/min critical to ±100 μm, and the root mean square (RMS) was calculated for the whole deviation. RMS can be expressed as follows [28,29,32]:(1)$$\mathrm{RMS}=\sqrt{\frac{\sum_{i=1}^{n}{\left(x_{1,i}-x_{2,i}\right)}^{2}}{n}},$$
where *x*_1,*i*_ are reference data, *x*_2,*i*_ are scan data, and *n* indicates the total number of measurement points measured in each analysis.

### 2.6. Statistical Analysis

Obtained data were analyzed with a statistical program (IBM SPSS Statistics 23; IBM SPSS Inc., Armonk, NY, USA). Although Shapiro–Wilk and Kolmogorov–Smirnov tests were performed, they did not show normal distribution (*p* < 0.05). Thus, a non-parametric test, i.e., the Kruskal–Wallis H test, was used. In addition, the Mann–Whitney U test was implemented by applying the level of significance (0.05/3 = 0.016), adjusted by post-analysis Bonferroni correction. Type I-error was set to α = 0.05.

## 3. Results

The trueness of the ceramic crowns in quantitative analysis is shown in Table 3 and Table 4. The outer surface measurements for each group were as follows: AM, 38.30 ± 4.20 μm; IPS, 34.89 ± 4.74 μm; and CEL 40.38 ± 3.32 μm, whereas those of the inner surfaces were as follows: AM, 58.76 ± 6.55 μm; IPS 59.42 ± 8.89 μm; and CEL, 67.12 ± 3.76 μm (Table 3).

The trueness values of the internal parts for each group were as follows: AM, 50.20 ± 7.37 μm; IPS, 41.32 ± 4.42 μm; and CEL, 54.38 ± 3.72 μm; while those of the external parts were: AM 85.73 ± 19.3 μm; IPS 108.11 ± 12.9 μm; and CEL 103.34 ± 12.40 μm (Table 4).

Statistically significant differences were observed among the three groups on the outer surface (*p* < 0.05). However, there was no statistically significant difference between the AM and IPS groups and AM and CEL groups (*p* > 0.05) (Table 3). Although statistically significant differences were observed among the three groups on the inner surface (*p* < 0.05), no statistically significant differences were observed between the AM and IPS groups and IPS and CEL groups (*p* > 0.05) (Table 3). Significant differences were also observed among the three groups with respect to the internal part (*p* < 0.05), while no statistically significant difference was observed between the AM and CEL group (*p* > 0.05). Regarding the trueness of the external part measurement, statistically significant differences were found among the three groups (*p* < 0.05). However, there was no statistically significant difference between the AM and CEL and IPS and CEL groups (*p* > 0.05).

The qualitative data of the outer and inner surfaces of the crowns were compared using a color difference map, where green denotes intolerance range, blue denotes negative error, and red denotes positive error (Figure 3). First, positive errors were observed in distal and mesial fossa of occlusal areas on the outer surfaces of AM A, IPS B, and CEL C. Second, the inner surface showed negative errors in AM D, IPS E, and CEL F occlusal areas, while positive errors were observed in the axial areas (Figure 3).

As shown in Figure 4, the inner surface was classified into internal and external parts using a color difference map, and trueness was compared between the ceramic crowns. Third, axial area errors were visible in the measurements of the internal parts of AM A, IPS B, and CEL C crowns. In particular, the CEL group showed wider positive errors than the AM and IPS groups. The degree of error in the IPS group was smaller than that in the other groups. Most of the external parts of the AM D, IPS E, and CEL F revealed green tolerance scopes, whereas the negative errors for the outer external areas in the IPS and CEL groups were more widely distributed than those in the AM group.

## 4. Discussion

As the dental CAD/CAM system is now widely used in dentistry, digital-based change has occurred in the diagnosis and crown production. Measuring the accuracy of the produced crowns using CAD/CAM-only ceramics is meaningful from the viewpoint of assessing their clinical applicability.

Previous studies have mainly evaluated the quality of prostheses through limited two-dimensional (2D) analysis on marginal fits of protheses [4,5,6,7,8]. However, recent studies have shown that process errors can be analyzed by conducting protheses 3D analysis [6,13,26,27,28,29,30]. A 2D analysis evaluates marginal discrepancy between prothesis and teeth, and lowers the experimental accuracy depending on the participants, in addition to being time-consuming. Conversely, the 3D measurement can access all parts, including undercut parts that are difficult to measure, and provides higher accuracy as it analyzes data through 3D visualization of a prothesis through scanning. Therefore, this study qualitatively and quantitatively analyzed the milling accuracy of lithium disilicate and zirconia-reinforced silicate crowns, fabricated using a dental CAD/CAM system, through 3D analysis.

The null hypothesis of this study is that there is no significant difference between the milling accuracy of lithium disilicate and zirconia-reinforced silicate crown fabricated using the dental CAD/CAM system. Based on the results obtained, the null hypothesis was rejected (*p* < 0.05).

In this study, the final fabricated ceramic crowns were defined by the outer and inner surfaces and internal and external parts. While the outer surface plays a crucial role in the occlusal force and optimal esthetics, the inner surface affects the life and fit of the prosthesis [33]. To thoroughly analyze the milling errors that appear on the inner surface of the ceramic crown, we divided the internal and external parts of the crown based on a 1 mm gap above the crown margin. Studies have reported that if the milling accuracy of the inner surface is not accurate, the life of the prosthesis may be shortened, and secondary caries may occur in the restored tooth, thus necessitating an in-depth analysis [13,14,15].

As shown in Table 3, statistically significant differences were presented, both on the outer and inner surface (*p* < 0.05). In both cases, the RMS values of the CEL group were high. In contrast, the AM and IPS groups showed the lowest values, and no significant difference was observed between two groups (*p* > 0.05). The higher the RMS value, the more the inaccuracy, while the lower the RMS value, the better the milling accuracy.

The highest value obtained in the CEL group could be attributed to the presence of 10% zirconia in the lithium silicate content, thus increasing the strength. A previous study showed that tangential and normal forces for zirconia-reinforced lithium silicate were approximately 10–30% higher than those for lithium disilicate, especially at higher specific removal rates of 1.35–1.8 mm^3^/mm/min [34]. Moreover, grinding force is largely influenced by material mechanical properties, as high-strength materials generally require high grinding force, and it is also significantly influenced by the applied processing conditions and tools that determine the material removal rate [34].

Another study calculated machinability outcome on CAD/CAM ceramic materials. Lithium disilicate showed a machinability of 12 ± 0.46 mm/min, and zirconia-reinforced silicate showed a machinability of 0.80 ± 0.21 mm/min, thus suggesting that lithium disilicate had better machinability than zirconia-reinforced silicate [22]. Other studies suggest that the mechanical properties of ceramic materials were related to the availability for milling and machinability and can have adverse impacts on complete crystallization of materials [22,34]. The results from this study are in line with the previously reported studies.

In this study, through qualitative analysis, the common milling error of the outer and inner surfaces and milling error according to the difference in ceramic materials were analyzed (Figure 3). In the qualitative analysis, positive and negative errors were expressed, with positive errors in red and yellow and negative errors in sky blue and blue. A positive error indicates a less-milled portion when milling a dental prosthesis, whereas a negative error indicates an excessively milled portion.

First, common milling errors of the outer surfaces of all ceramic crowns were found to be positive errors in the mesial and distal fossa in the occlusal area (Figure 3A–C). In particular, the distal fossa of the outer surface showed a severe red positive error because the grinding bur used in the milling machine had a wide and round shape. Therefore, the micro-reproducibility was poor in narrow areas such as the fossa [24]. However, only slight differences were found when different groups of ceramic materials were compared through qualitative analysis (Figure 3).

Second, the common milling errors of all ceramic crowns on the inner surface were found in the axial area and occlusal area (Figure 3D–F). In particular, positive error was observed in the axial area because the inner surface was further cut during the processing of the crown. While the milling device was composed of the 3 + 1 axes used in this experiment, positive errors occurred, as the sophisticated fabrication was insufficient in the axial area during the grinding process (Figure 3 and Figure 4). In contrast, negative errors occurred in the occlusal area (Figure 3 and Figure 4). It is a normal phenomenon in the CAD/CAM system for a similar reason identified in a previous study [6,35]. Due to this, marginal discrepancy can be wide in the occlusal area.

When the outer and inner surfaces of all the ceramic crowns were qualitatively analyzed, it was not easy to analyze significant differences (unlike the quantitative analysis presented in Table 3) on the outer surface. However, on the inner surface, it was possible to qualitatively analyze the difference in milling accuracy, depending on the type of ceramic (Figure 3 and Figure 4).

In the inner surface, the CEL group showed a wider extent of errors in the axial area compared to the AM and IPS groups (Figure 3D–F). Since 10% zirconia element was contained inside the glass ceramic and increasing bending strength corresponds to increasing brittleness value, AM and IPS showed superior machinability than the CEL group [18].

For an in-depth examination of the errors related to the inner surface, both internal and external parts were analyzed (Table 4) (Figure 4). The internal part is directly associated with cement space and absorption of masticatory pressure [33], whereas the external part is significant as it can trigger secondary caries and periodontal disease caused by external bacterial invasion [13]. In quantitative analysis, the internal part showed the result values in the order of IPS, AM, and CEL group (Table 4), and a statistically significant difference was found between the AM and IPS and IPS and CEL groups (*p* < 0.05). In the AM and IPS groups, a difference in the results could be due to the difference in the content of the basic ingredients between the same lithium disilicate materials or manufacturers.

However, when the internal parts of all ceramic crowns were observed by qualitative analysis, the axial area of the CEL group showed a wider range of positive errors than the lithium disilicate AM and IPS groups, and the distribution of milling errors was wide (Figure 4A–C). The IPS and AM groups showed small process errors compared to the CEL group (Figure 4A–C).

Similar results were observed on the external part (Table 4). Values were smallest in the AM group, followed by the CEL group, and then the IPS group (Table 4). Although a significant difference was found between the AM and IPS groups (*p* < 0.05), no significant difference was noted between the AM and CEL and IPS and CEL groups (*p* > 0.05). In particular, comparing the two results through qualitative analysis of the external part, the AM group showed smaller negative errors compared to the CEL group (Figure 4D–F).

The positive error appearing in the internal and external parts may not sufficiently match the fitting of the prosthesis, resulting in micro-discrepancies, and a negative error may loosen the prosthesis due to excessive deletion. However, as shown in Table 3 and Table 4, the clinical stability criterion was within 120 μm based on fitness [36], and the errors resulting from 3D milling were within 120 μm. Therefore, these are considered to be appropriate for clinical use.

Overall, this study demonstrated that ceramic restoration with a CAD/CAM system is influenced by brittleness or chipping as the material mechanical properties. Marginal fit or internal fit may be affected by the type of ceramics. In short, it is assumed that the number of processing errors in the course of fabricating ceramic restoration was higher when using zirconia-reinforced silicate than lithium disilicate. Therefore, this study recommends lithium disilicate for fabricating ceramic restorations over zirconia-reinforced silicate.

A limitation of this study lies in limited teeth models. Despite different preparation shapes, such as inlay or onlay, this study selected only limited crowns. In addition, the processability of crowns can vary according to teeth shapes. Thus, further studies are required to produce ceramic restorations using a diverse range of ceramics and crown shapes, including inlay or onlay, and again measure the milling accuracy and compare the clinical values.

## 5. Conclusions

Notwithstanding the limitations of this in vitro study, the following conclusions can be drawn:After processing of ceramic restorations using a CAD/CAM system, lithium disilicate was shown to have superior milling accuracy compared to zirconia-reinforced lithium silicate.According to the results from the qualitative analysis, a positive error appearing in the internal and external parts of the crown may not sufficiently match the fitting of the prosthesis, resulting in micro-discrepancies. In addition, a negative error may loosen the prosthesis due to excessive deletion.According to the results from the quantitative analysis, milling accuracy was within 120 μm for all types of ceramics, thus confirming their clinical applicability.

## Figures and Tables

**Figure 1 materials-13-04680-f001:**
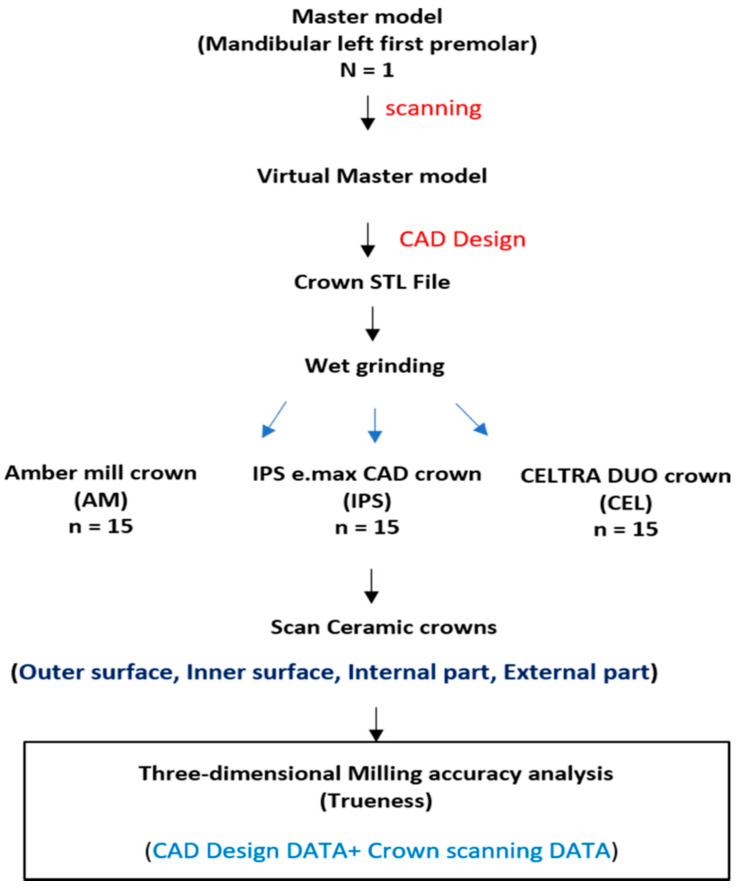
Diagram of the experimental procedure.

**Figure 2 materials-13-04680-f002:**
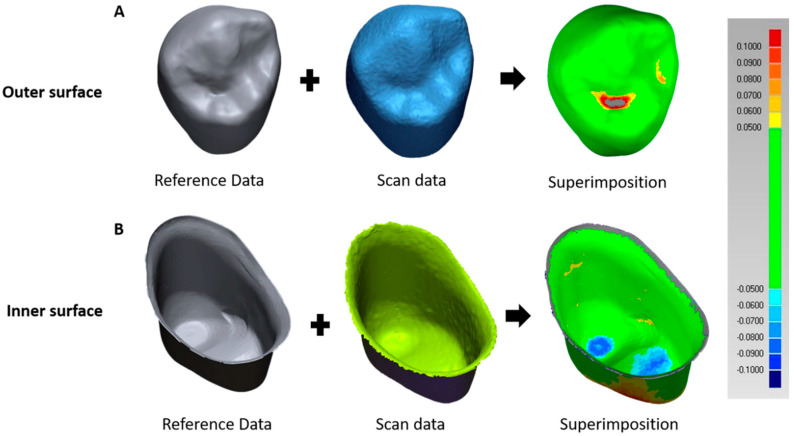
Milling accuracy evaluation method using a three-dimensional program. (**A**) Superimposing of outer surface reference data and scan data using a three-dimensional program. (**B**) Superimposing of inner surface reference data and scan data using a three-dimensional program.

**Figure 3 materials-13-04680-f003:**
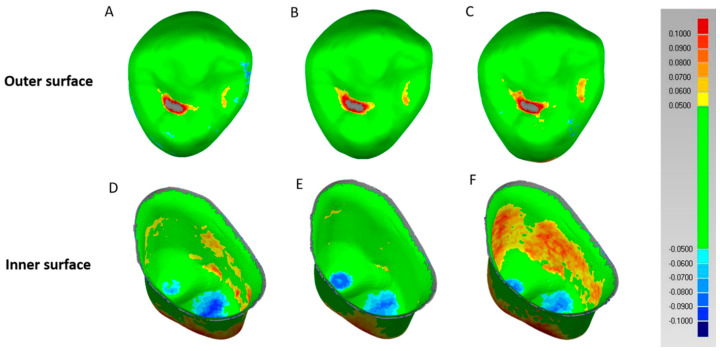
Evaluation of the trueness of crowns, prepared using the three types of ceramic blocks, in the outer and inner surfaces. (**A**) Color difference in the outer surface of the AM crown, showing the superimposing of 3D data. (**B**) Color difference in the outer surface of the IPS crown, showing the superimposing of 3D data. (**C**) Color difference in the outer surface of the CEL crown, showing the superimposing of 3D data. (**D**) Color difference in the inner surface of the AM crown, showing the superimposing of 3D data. (**E**) Color difference in the inner surface of the IPS crown, showing the superimposing of 3D data. (**F**) Color difference in the inner surface of the CEL crown, showing the superimposing of 3D data.

**Figure 4 materials-13-04680-f004:**
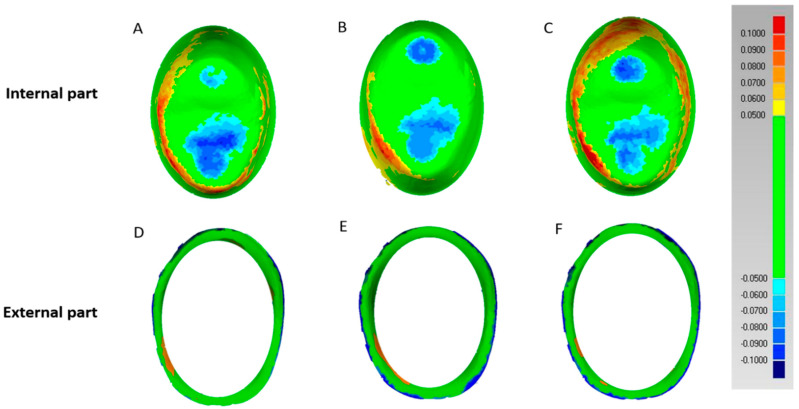
Evaluation of the trueness of crown for the three types of ceramic blocks in the internal and external parts. (**A**) Color difference in the internal part of the AM crown, showing the superimposing of 3D data. (**B**) Color difference in the internal part of the IPS crown, showing the superimposing of 3D data. (**C**) Color difference in the internal part of the CEL crown, showing the superimposing of 3D data. (**D**) Color difference in the external part of the AM crown, showing the superimposing of 3D data. (**E**) Color difference in the external part of the IPS crown, showing the superimposing of 3D data. (**F**) Color difference in the external part of the CEL crown, showing the superimposing of 3D data.

**Table 1 materials-13-04680-t001:** Base ceramic compositions.

Group	Materials	Shade	Basic Composition	Manufacturer
AM	Lithium disilicate	HT A2	SiO_2_, Li_2_O, P_2_O_5_, Al_2_O_3_ other oxides and colorants	HASS Corp
IPS	Lithium disilicate	HT A2	SiO_2_, Li_2_O, K_2_O, MgO, Al_2_O_3_, P_2_O_5_ other oxides	Ivoclar Vivadent AG
CEL	Zirconia-reinforced lithium silicate	HT A2	SiO_2_, LiO_2_, ZrO_2_, P_2_O_5_, Al_2_O_3_, K_2_O, CeO_2_, other oxides	Sirona Dentsply

**Table 2 materials-13-04680-t002:** Heat-treatment schedule of ceramics in furnace.

Group	B (°C)	S (min)	t1 (°C/min)	T (min)	H (min)	V1 (°C)	V2 (°C)	L (°C)
AM	400	6.00	60	810	15.00	550	810	680
IPS	403	6.00	90	830	10.00	550	830	710
CEL	500	3:30	60	820	1:00	off	off	750

B: stand by temperature; S: closing time; t1: temperature rate increase; T: holding temperature; H: holding time; V1: vacuum on temperature; V2: vacuum off temperature; L: long-term cooling temperature.

**Table 3 materials-13-04680-t003:** Results of trueness Root Mean Square (RMS) for outer and inner surfaces of the ceramic crowns.

Trueness Results (RMS) for the Outer and Inner Surface of the Ceramic Crowns						
Group	Outer Surface			Inner Surface		
	Mean ± SD	95% CI	*p*-Value	Mean ± SD	95% CI	*p*-Value
AM	38.30 ± 4.20 ^a,b^	35.97–40.63	<0.001	58.76 ± 6.55 ^a^	55.13–62.38	<0.016
IPS	34.89 ± 4.74 ^a^	32.26–37.51		59.42 ± 8.89 ^a,b^	54.49–64.34	
CEL	40.38 ± 3.32 ^b^	38.54–42.21		67.12 ± 3.76 ^b^	65.03–69.20	

Unit: μm. AM: Amber Mill, CEL: CELTRA DUO, IPS: IPS e.max CAD, CI: confidence interval, SD: standard deviation. ^a,b^ values followed by statistically significant differences based on the Mann–Whitney U test with Bonferroni correction (*p* < 0.05).

**Table 4 materials-13-04680-t004:** Results of trueness Root Mean Square (RMS) for internal and external parts of the ceramic crowns.

Trueness Results (RMS) for the Internal and External Part of the Ceramic Crowns						
Group	Internal Part			External Part		
	Mean ± SD	95% CI	*p*-Value	Mean ± SD	95% CI	*p*-Value
AM	50.20 ± 7.37 ^a^	46.12–54.28	<0.001	85.73 ± 19.30 ^a^	75.05–96.41	<0.006
IPS	41.32 ± 4.42 ^b^	38.87–43.76		108.11 ± 12.94 ^b^	100.9–115.3	
CEL	54.38 ± 3.72 ^a^	52.32–56.45		103.34 ± 12.40 ^ab^	96.48–110.2	

Unit: μm. AM: Amber Mill, CEL: CELTRA DUO, IPS: IPS e.max CAD, CI: confidence interval, SD: standard deviation. ^a,b^ values followed by statistically significant differences based on the Mann–Whitney U test with Bonferroni correction (*p* < 0.05).

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
