# Peer review of "Evaluation of the Milling Accuracy of Zirconia-Reinforced Lithium Silicate Crowns Fabricated Using the Dental Medical Device System: A Three-Dimensional Analysis"

_materials, 2020, doi:10.3390/ma13204680_

Round 1
Reviewer 1 Report
"Evaluation of the milling accuracy of zirconia-reinforced lithium silicate crowns fabricated using dental medical device system: A three-dimensional analysis”
It is very interest to focus on comparing the milling accuracy of lithium disilicate and zirconia reinforced silicate crowns fabricated using the dental CAD / CAM system. This paper is very well drawn. Please refer to the following comments.
minor
1) Please align the order of Amber Mill, CELTRA DUO and IPS e.max CAD in Figure 1 with the same order as the other Tables. Also, the resolution is poor and the characters are blurry, so please correct it.
2) The text on lines 213-217 is ambiguous. Please pay attention to the English grammar and correct it with the author's expression.
3)The definitions of a and b are incorrect in the explanations of Tables 3 and 4. Please define it exactly.
Author Response
Response to Reviewer 1 Comments
It is very interest to focus on comparing the milling accuracy of lithium disilicate and zirconia reinforced silicate crowns fabricated using the dental CAD / CAM system. This paper is very well drawn. Please refer to the following comments.
Thank you for the positive feedback. Based on the comments/suggestions provided by the reviewers, we have further improved the quality of the paper.
Point 1: Please align the order of Amber Mill, CELTRA DUO and IPS e.max CAD in Figure 1 with the same order as the other Tables. Also, the resolution is poor and the characters are blurry, so please correct it.
Response 1: Thank you for the helpful suggestions. We have made the necessary changes in the revised manuscript (P.3-82, P.5- 122, P.7- 173, P.8- 191).
Point 2: The text on lines 213-217 is ambiguous. Please pay attention to the English grammar and correct it with the author's expression.
Response 2: Thank you for the valuable suggestion. We appreciate your helpful advice and have revised the text accordingly.
- To thoroughly analyze the milling errors that appear on the inner surface of the ceramic crown, we divided the internal and external parts of the crown based on a 1-mm gap above the crown margin. Studies have reported that if the milling accuracy of the inner surface is not accurate, the life of the prosthesis may be shortened, and secondary caries may occur in the restored tooth; thus necessitating an in-depth analysis [13, 14, 15]. (P.10, 219-224)
Point 3: The definitions of a and b are incorrect in the explanations of Tables 3 and 4. Please define it exactly.
Response 3: Thank you for pointing out this discrepancy. As suggested, we have corrected the text accordingly (P.6, 146, 153).
- a, b values followed by statistically significant differences based on the Mann-Whitney U test with Bonferroni correction (P < 0.05).

Reviewer 2 Report
Dear Authors,
congratulations for your work which I found very interesting. I just have some minor revisions to propose to you in order to improve your work.
INTRODUCTION
Line 48-49: I think you should stress the attention on the fact that every effort should be done to prevent decay, not only as regards prostheses but also restorative dentistry. As a reference you could cite the following article: Colombo M, Gallo S, Poggio C, Ricaldone V, Arciola CR, Scribante A. New Resin-Based Bulk-Fill Composites: in vitro Evaluation of Micro-Hardness and Depth of Cure as Infection Risk Indexes. Materials (Basel). 2020 Mar 13;13(6):1308.
Line 73: I suggest rephrasing the sentence such as “The null hypothesis of this study is that there is no significant difference between…..”
MATERIALS AND METHODS
Line 81: please add a reference for ISO 554. Is it reference n°24? Please do not insert references with superscript numbers.
Line 102-103: please rephrase.
Line 112: please substitute “seoul” with “Seoul”.
DISCUSSION
Line 204: I suggest using “Conversely” instead of “However”.
Line 209-210: please rephrase as for line 73.
Line 215: I think the sentence is suspended.
Thank you for your work and congratulations again.
Author Response
Response to Reviewer 2 Comments
congratulations for your work which I found very interesting. I just have some minor revisions to propose to you in order to improve your work.
Thank you for the positive and encouraging feedback. We have further improved the quality of the paper based on the helpful suggestions provided by the reviewers.
Point 1: INTRODUCTION
Line 48-49: I think you should stress the attention on the fact that every effort should be done to prevent decay, not only as regards prostheses but also restorative dentistry. As a reference you could cite the following article: Colombo M, Gallo S, Poggio C, Ricaldone V, Arciola CR, Scribante A. New Resin-Based Bulk-Fill Composites: in vitro Evaluation of Micro-Hardness and Depth of Cure as Infection Risk Indexes. Materials (Basel). 2020 Mar 13;13(6):1308
Response 1: Thank you for the valuable suggestion. We appreciate your helpful inputs and have added the following text in the revised manuscript.
- When these prostheses are used for fabricating ceramic restoration, there are chances of clinical failure due to the absorption of occlusal force. Furthermore, dental caries and periodontal disease can be triggered due to prosthesis fracture, microleakage, or external bacterial invasion [13, 14, 15]. Indeed, it is important to manufacture a perfect prosthesis in order to avoid problems that may occur during the restoration of the dental prosthesis in the oral cavity. However, the most reliable method to protect the teeth is to prevent the initial formation of cavities in the oral cavity [16]. (P.2, 46-52)
Line 73: I suggest rephrasing the sentence such as “The null hypothesis of this study is that there is no significant difference between…..”
Response 2: Thank you for the valuable input. We have rephrased the sentence based on your suggestion as follows:
- The null hypothesis of this study is that there is no significant difference between the milling accuracy of lithium disilicate and zirconia-reinforced silicate crown, fabricated using the dental CAD/CAM system. (P.2, 76-78)
Point 2: MATERIALS AND METHODS
Line 81: please add a reference for ISO 554. Is it reference n°24? Please do not insert references with superscript numbers.
Response 3: Thank you for the valuable input. We have rephrased the sentence based on your suggestion as follows: p. 3(84)
- The experiment was conducted at a temperature of 23 ± 1 °C, in accordance with ISO 554 [31].
- 30. International Organization for Standardization (ISO). Standard Atmospheres for Conditioning and/or Testing -Specifications; ISO 554: 1976; ISO: Geneva, Switzerland, 1976.
Line 102-103: please rephrase.
Response 4: Thank you for the valuable input. We have rephrased the sentence based on your suggestion as follows: p. 4(105-107)
- After milling was completed, completed ceramic crowns attached to the ceramic blank were separated by removing the connected holder, Remaining parts were removed using a diamond bur grinder.
Line 112: please substitute “seoul” with “Seoul”.
Response 5: Thank you for the valuable input. We have rephrased the sentence based on your suggestion as follows: p. 4(116)
- lab scanner (Identica blue, Medit, Seoul, Korea).
Point 3: DISCUSSION
Line 204: I suggest using “Conversely” instead of “However”.
Response 6: Thank you for the suggestion. We have made the necessary change in the revised manuscript (P.8, 210).
Line 209-210: please rephrase as for line 73.
Response 7: Thank you for the suggestion. We have corrected the text accordingly (P.8, 215-217).
Line 215: I think the sentence is suspended.
Response 8: Thank you for pointing this out. We have corrected the text accordingly.
(P.10, 220-225).
- To thoroughly analyze the milling errors that appear on the inner surface of the ceramic crown, we divided the internal and external parts of the crown based on a 1-mm gap above the crown margin. Studies have reported that if the milling accuracy of the inner surface is not accurate, the life of the prosthesis may be shortened, and secondary caries may occur in the restored tooth; thus necessitating an in-depth analysis [13, 14, 15] (P.10, 220-225).

Reviewer 3 Report
120, 169, 187 Explanations to fig. 2, 3 and 4 should be included in the caption.
e.g. Figure 4. Evaluation of the trueness of crown for the three types of ceramic blocks, in the internal and external parts. Color difference in the internal part of A (AM), B (IPS), C (CEL) and external D (AM), E (IPS), F (CEL) crowns, showing the superimposing of 3D data.
141 Table 3 should not be split.
105, 141 and 148 Tables are unreadable, they should be formatted differently.
Text from 283 to 293 should be included in the conclusions.
Author Response
Response to Reviewer 3 Comments
Thank you for the positive and encouraging feedback. We have further improved the quality of the paper based on the helpful suggestions provided by the reviewers.
Point 1: 120, 169, 187 Explanations to fig. 2, 3 and 4 should be included in the caption.
e.g. Figure 4. Evaluation of the trueness of crown for the three types of ceramic blocks, in the internal and external parts. Color difference in the internal part of A (AM), B (IPS), C (CEL) and external D (AM), E (IPS), F (CEL) crowns, showing the superimposing of 3D data.
Response 1: Thank you for the valuable suggestion. We have made the necessary changes in the revised manuscript. P.5 (124-127), P.7 (175-182), P.8 (193-200)
Point 2: 141 Table 3 should not be split.
Response 2: Thank you for the suggestion. We have revised Table 3 accordingly.
Point 3: 105, 141 and 148 Tables are unreadable, they should be formatted differently.
Response 3: Thank you for providing this input. Based on the suggestion, we have reformatted the relevant tables. (P. 4, 104-112, P. 6, 145-157)
Point 4: Text from 283 to 293 should be included in the conclusions.
Response 4: Thank you for the helpful suggestion. We have corrected the text accordingly. (P. 10, 303-314)
- Notwithstanding the limitations of this in vitro study, the following conclusions can be drawn:
- After processing of ceramic restorations using a CAD/CAM system, lithium disilicate was shown to have superior milling accuracy compared to zirconia-reinforced lithium silicate.
- Ceramic restoration with a CAD/CAM system is influenced by brittleness or chipping as the hardness or strength increases. Marginal fit or internal fit may be affected by the type of ceramic used.
- According to the results from the qualitative analysis, a positive error appearing in the internal and external parts of the crown may not sufficiently match the fitting of the prosthesis, resulting in micro discrepancies. In addition, a negative error may loosen the prosthesis due to excessive deletion.
- According to the results from the quantitative analysis, milling accuracy was within 120 μm for all types of ceramics, thus confirming their clinical applicability.
